# From healthy to unhealthy obesity: A longitudinal study of adults in ELSA-Brasil

**Fernanda Duarte Mendes[1], Hully Cantão dos Santos[2], José Geraldo Mill[2], Maria Del Carmen Bisi Molina[2], Maria de Fátima H. Sander Diniz[3], Carla Romagnolli Quintino[4], Márcio Sommer Bittencourt[5], Carolina Perim de Faria[1]**

1 Post-Graduate Program in Nutrition and Health, Health Sciences Center, Federal University of Espírito Santo, Vitória, Espírito Santo, Brazil, 2 Post-Graduate Program in Public Health, Health Sciences Center, Federal University of Espírito Santo, Vitória, Espírito Santo, Brazil, 3 School of Medicine, Federal University of Minas Gerais, Belo Horizonte, Minas Gerais, Brazil, 4 Department of Internal Medicine, University of São Paulo, São Paulo, São Paulo, Brazil, 5 School of Medicine, University of Pittsburgh, Pittsburgh, Pennsylvania, United States of America

\* carolina.faria@ufes.br

## Abstract

Despite obesity being associated with negative metabolic and cardiovascular outcomes, there is a subgroup of individuals considered healthy. However, there are questions about the stability of the Metabolically Healthy Obesity phenotype. This is a longitudinal study using the ELSA-Brasil cohort, conducted from 2008/10–2017/19 aiming to describe the trajectory of metabolic status of individuals with obesity, as well as the factors associated with the transition into the unhealthy status. Metabolic status was determined using measures of blood pressure, fasting glucose/glycated hemoglobin, triglycerides, and HDL-cholesterol, no previous diagnosis of alteration in any of these parameters nor taking medication to control them. SPSS v.21.0 was used, considering p < 0.05 as significant. The sample consisted of 190 Metabolically Healthy Individuals with Obesity at baseline, of whom 75.8% transitioned to Metabolically Unhealthy status on the third wave of the study. The baseline data indicates that 8.6% of individuals with obesity were metabolically healthy, and in the follow-up, the prevalence was 5.5%. Alcohol use was a risk factor for metabolic status transition [RR: 1.359 (95%CI: 1.005–1.838)]. Also, each 1 cm increase in waist circumference contributed to a 1% increase in the risk of transitioning from healthy to unhealthy metabolic status [RR: 1.011 (95%CI: 1.004–1.018)]. Being a metabolically healthy individual with obesity is a transient state and alcohol consumption as well as increases in waist circumference are risk factors for the metabolic transition.

## Introduction

The prevalence of obesity has been rising worldwide, and because of this epidemic, there has also been an increase in the incidence of noncommunicable chronic

**Data availability statement:** Due to ethical restrictions approved by the ethics committee of each institution (Universidade Federal de Minas Gerais, Universidade de São Paulo, Universidade Federal do Espírito Santo, Universidade Federal do Rio Grande do Sul, Universidade Federal da Bahia e Fundação Oswaldo Cruz) and by the Publications Committee of ELSA-Brasil (publiELSA), the data used in this study can be made available for research proposals by a request to ELSA's Datacenter (rb.sgrfu@asleacitsitatse) and to the ELSA's Publications Committee. Additional information can be obtained from the ELSA Coordinator from the Research Center of Espírito Santo (jose.mill@gmail.com).

**Funding:** This research was funded by the Brazilian Ministry of Health (DECIT—Department of Science and Technology) and Ministry of Science and Technology (FINEP—Research Funding Agency and CNPq—National Council for Scientific and Technological Development—process numbers 01 06 0010.00 RS, 01 06 0212.00 BA, 01 06 0300.00 ES, 01 06 0278.00 MG, 01 06 0115.00 SP, 01 06 0071.00 RJ). These sources had no role in study design or publication.

**Competing interests:** The authors have declared that no competing interests exist.

diseases [1–3]. This occurs due to the chronic inflammatory state caused by obesity, as well as the influence that excess fat has on various unfavorable cardiometabolic outcomes [3,4]. However, the literature shows that not all individuals with obesity exhibit the same cardiometabolic risk [5,6]. It is believed that there is a subset of individuals with obesity who can be considered metabolically healthy [5,7].

The phenotype of metabolically healthy obesity (MHO) refers to individuals with obesity who do not show alterations in cardiometabolic markers [5]. These individuals have normal values of blood pressure (BP), fasting glucose, HDL cholesterol, triglycerides (TG), waist circumference (WC), and other measures [1,8,9]. Additionally, they do not use medications for dyslipidemia, hypertension, or diabetes, and have no presence of cardiovascular disease [8,9]. Although this topic has recently been more explored, our understanding of MHO is still limited [10]. The scientific literature is still divided on the topic. While some authors suggest that MHO exists and can be considered a protective state against cardiometabolic alterations and mortality, others argue that MHO is merely a stage of metabolic balance, marking the transition between a healthy state and the classification of metabolically unhealthy obesity (MUO) [1,2].

Starting from the assertion that MHO is a transient state, questions arise about whether it is indeed an unstable condition, and which variables might be associated with the transition from MHO to MUO [11]. The literature indicates sex and age as related variables, sedentary behaviour, smoking, and continuous alcohol use also appear to be risk factors for transitioning to an unhealthy status [12,13]. Additionally, dietary intake, particularly the consumption of ultraprocessed foods, has been associated with an unfavorable metabolic profile [14].

Thus, our objective is to assess whether metabolically healthy obesity is a transient state, as well as to identify the factors that influence the transition to metabolically unhealthy obesity. The study analyzes the trajectory of metabolic status and associated variables from baseline and the third wave of follow-up in the ELSA-Brasil cohort.

## Methods

This is a longitudinal study conducted using the baseline or wave 1 (W1) (August 2nd 2008- December 17th 2010) and the third wave (W3) (April 2nd 2017- May 31st 2019) of the Brazilian Longitudinal Study of Adult Health (ELSA-Brasil). ELSA-Brasil is a multicenter cohort study of 15,105 adults (35–74 years old at baseline, 2008–2010), all active or retired civil servants of six Brazilian public universities and one research institution. Detailed information about the cohort has been published elsewhere [15].

The sample is composed of all participants classified as MHO at baseline. Those who did not have data regarding the MHO diagnostic variables at baseline and follow-up were excluded from this analysis. Individuals were classified as having MHO based on the criteria proposed by Quintino [16]. According to these criteria, individuals are considered healthy if they do not meet any of the conditions listed in Table 1.

Body weight, height, and waist circumference (WC) were collected according to the methods proposed by Lohman et al. [17]. BMI was calculated and categorized

**Table 1. Criteria for evaluating metabolically healthy obesity (MHO) according to the cutoff points proposed by Quintino [2018].**

|  | CRITERIA |
|---|---|
| **Blood pressure abnormalities** | A self-reported medical diagnosis of arterial hypertension OR Antihypertensive drug use OR Systolic blood pressure ≥ 130 mmHg OR Diastolic blood pressure ≥ 85 mmHg |
| **Glucose metabolism abnormalities** | A self-reported medical diagnosis of diabetes mellitus OR Antidiabetic drugs use OR Fasting blood glucose ≥ 100mg/dL OR Glycated hemoglobin ≥ 6,5% |
| **Lipids metabolism abnormalities** | A self-reported medical diagnosis of hypertriglyceridemia OR Triglycerides ≥ 150 mg/dL OR use of statins<br>A self-reported medical diagnosis of low HDL OR HDL <40 mg/dL (men) or <50 mg/dL (women) OR use of statins |

Note: HDL: *High-density lipoprotein.*

according to WHO [18]: "Underweight" (BMI < 18.5 kg/m2), "Normal weight" (BMI 18.5 - 24.9 kg/m2), "Overweight" (BMI 25.0 - 29.9 kg/m2), and "Obesity" (BMI ≥ 30.0 kg/m2), considering for the study those individuals who were in the "obesity" category [15].

Blood collection was carried out after an average fast of 12 hours, with details described by Fedelli et al. (2013) [19].

Blood pressure (BP) was measured while subjects were still fasting, in a sitting position after a minimum of 5 min resting period in the left arm. Three measurements were obtained at one-minute intervals and the average of the last two measurements was considered as the casual BP.

All medications used regularly were obtained during an interview on the same day of the exams and they were coded according to the Anatomical Therapeutic Chemical classification. The presence of the following classes of medications was considered: antihypertensives, antidiabetic agents, and fibrates, niacins, and statins. The use of each of the registered drugs was expressed as a binary variable (yes/no) [15].

The assessment of individuals' habitual food consumption was conducted using the Food Frequency Questionnaire (FFQ). This is a semi-quantitative instrument designed to estimate dietary intake over the past twelve months. The FFQ applied was structured with food items/preparations, portion sizes, and consumption frequencies. It offers eight response options, ranging from "More than 3 times/day" to "Never/Almost Never." Seasonal consumption was also considered for individuals who reported consuming a specific food item only during a particular time of year or season.

At baseline, the FFQ contained 114 food items and was validated for the Brazilian population [20]. In wave 3, a shortened version of the FFQ was used, reducing the number of items from 114 to 76, a 33% reduction. However, the shortened version maintained good capacity to measure energy and nutrients and it was also validated for the population [21].

The Nutrition Data System for Research software was used to analyze the consumption data reported in the FFQ. Extreme consumption values (above the 99th percentile) were replaced with the exact value of the 99th percentile. Additionally, when participants voluntarily reported seasonal consumption of a specific item, the total daily intake of that food was multiplied by 0.25 [20,21].

For this study, the classification of foods according to the degree of processing was adopted. The study followed the NOVA classification proposed by Monteiro et al. (2016) [22] which separates foods into four groups: unprocessed or minimally processed foods, processed culinary ingredients (MPF), processed foods (PF), and ultra-processed foods (UPF) [14]. The next step estimated the contribution in calories and grams of each food category and its proportion considering daily food/calorie intake. After an initial analysis, the percentual contribution of each food category was chosen to be addressed in grams, as there was no significant difference between the two options and the variable in grams allows the inclusion of light/diet beverages in the study. Food intake was analyzed longitudinally (dynamic model), using average consumption data from baseline and follow-up. For the calculation of the variable, the contribution in grams of each food group was used, as well as the percentage contribution of each group to the diet. This calculation was made based on the dietary consumption data from baseline and wave 3. To generate the dynamic variable, the average of the values found at

both time points was calculated. Furthermore, the variable "dietary change" was utilized to assess the impact of a potential shift in dietary habits while maintaining metabolic status, and to justify the findings regarding the relationship between dietary consumption and metabolic status. The International Physical Activity Questionnaire (IPAQ) long version, validated for Brazil by Matsudo et al. [23] was used for the measurement of physical activity (PA) in minutes/week; it was done by multiplying the weekly frequency by the duration of each activity performed. The classification proposed by the WHO for insufficient, moderate, and vigorous activity was used, which recommends at least 150 minutes of moderate PA per week, or 75 minutes or more of vigorous PA [24,25].

Sociodemographic variables were collected by a standardized questionnaire administered in an interview. Sex was categorized as male and female and age into quartiles. Race/skin color was categorized as white, black, brown, and yellow or indigenous. Education was categorized as primary education, secondary education, and higher education. The occupational category variable classifies individuals according to the type of work and is categorized as higher, middle, and manual. Per capita family income was calculated based on the total net income of the family in Brazilian reais over the previous three months, divided by the number of people dependent on that income. The smoking variable was categorized as "former" "current," or "never" smoked. Alcohol consumption was measured using an Alcohol Consumption questionnaire, structured with close-ended questions, based on the National Center for Health Statistics questionnaire (1994) [26].

All variables are presented as proportions, means, and standard deviation (SD), or medians. The Kolmogorov-Smirnov test was used to assess the normality of data. For continuous variables, the student's t-test for independent samples and the Mann-Whitney test were used. For categorical variables, the chi-square test and Fisher's exact test were used.

The metabolic status transition between baseline and follow-up was dichotomized as No and Yes, where "No" indicates individuals who remained healthy in both study segments (baseline and follow-up), and "Yes" includes those who changed status from baseline to follow-up. Relative risk measures, both crude and adjusted, with their respective 95% confidence intervals were obtained through Poisson regression models with robust variance. All variables with $p < 0.20$ in the bivariate analysis were considered potential predictors of the metabolic status transition and inserted in the multivariate models. SPSS for Windows version 21 was used for statistical analyses adopting $p < 0.05$ as significant.

Sensitivity analyses were conducted to ensure greater precision of the findings. It was carried out in relation to the MHO diagnostic criterion, maintenance of bariatric participants, classification of foods items and the approach used in the analysis (grams or kcal) as well as assessing the diet at baseline, at follow-up, and the average of both and the information about diet changes over the previous 6 months and finally in relation to the variables of physical activity (PA). Final stage of quality control was an analysis of losses aiming to explore possible differences between the sample selected for the study and study losses and exclusions.

## Ethical considerations

The ELSA-Brasil protocol was approved by the National Ethics Committee under the registration number 140/08 and by the ethics committees of each participating CI, with the following registration numbers: 669/06 (USP), 343/06 (FIOCRUZ), 041/06 (UFES), 186/06 (UFMG), 194/06 (UFRGS), 027/06 (UFBA). As an inclusion criterion for the study, all those who wished to participate in the research read and signed the Informed Consent Form (ICF). The use of the data from this research was only possible after prior approval from the ELSA-Brasil's Project Publications Committee.

## Results

From the 15,105 participants included in the baseline of the ELSA-Brasil study. The average follow-up time was 7.7 years (minimum: 7 and maximum: 9). 3,540 of the participants (23,4%) were classified as individuals with obesity, having a mean BMI of 33.6 (SD) kg/m². 3,428 (96.8%) had all data necessary to obtain the metabolic classification and 294 (8.6%) were classified as presenting MHO. Fig 1 shows the flowchart, indicating the metabolic status of the participants at each of the two waves and the inclusion flow in the study from baseline. At follow-up, of the 3,084 individuals with obesity

evaluated, 169 (5.5%) were metabolically healthy. From those 294 MHO on baseline, 104 were lost due to follow-up or missing key variables, of these, 63.5% did not have the necessary data for the diagnosis of MHO, 30.7% refused to participate in the follow-up, 4.8% passed away, and 1% could not be located, resulting on a sample of 190 individuals that were included.

Table 2 shows that, at baseline, significant differences were found between groups concerning age (p<0.001) and sex (p<0.001), with women comprising the majority of those classified as MHO (78.6%). Married individuals were more prevalent among the unhealthy status (p=0.011), while those with higher education were more likely to be in the MHO group (52.4%, p<0.001). Manual occupation was more common among the MUO participants (p=0.007), and non-smoking was predominant in the MHO group (64.6%, p=0.001). Bariatric surgery was more prevalent in the MHO group (6.2%) compared to the MUO group (1.2%, p<0.001). Healthier individuals had lower BMI and waist circumference (WC) (p<0.001 for all). Processed food consumption was higher among MHO individuals (p=0.002).

In Table 3, the characterization of the metabolic status transition is described according to sociodemographic, lifestyle, anthropometric, and dietary consumption variables after an average 7.7 years of follow-up. When analyzing the status transition in follow-up, out of the 190 individuals classified as MHO at baseline and with all data in follow-up, 75.8% experienced status transition, moving from MHO to MUO. When comparing the group that transitioned from healthy to unhealthy status, statistically significant differences are noted in alcohol use (p=0.007), with alcohol consumers predominantly allocated in the "Yes" transition group (63.9%). Additionally, statistically significant data is observed for the variables of WC and waist-to-hip ratio, with lower mean values observed in individuals classified as "No," who did not experience a status transition (p=0.002 and p=0.001, respectively).

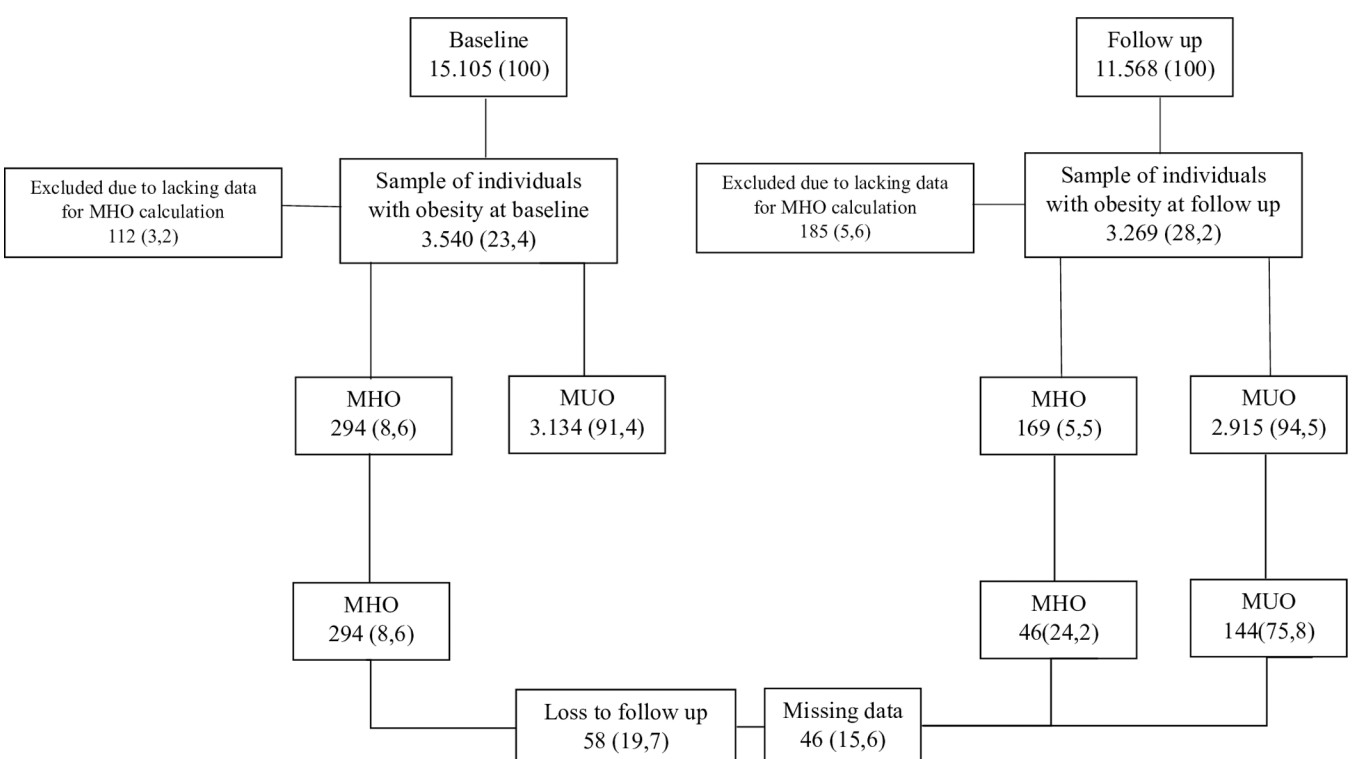

**Fig 1. Sample selection flowchart.** Note: Number indicates the number of subjects and in parenthesis percentage.

**Table 2. Pointwise characterization of metabolic status according to sociodemographic, lifestyle, anthropometric, and dietary intake variables at baseline and follow-up of the ELSA-Brasil study.**

| Variables | Baseline | | |
|---|---|---|---|
| | MHO | MUO | p-Value |
| | 294 (8,6) | 3.134 (91,4) | |
| **Age** | | | p<0,001* |
| Quartile 1(≤ 46.0 years) | 132 (44,9) | 765 (24,4) | |
| Quartile 2(47.0–52.0 years) | 83 (28,2) | 783 (25,0) | |
| Quartile 3(53.0–59.0 years) | 49 (16,7) | 849 (27,1) | |
| Quartile 4(≥ 60.0 years) | 30 (10,2) | 737 (23,5) | |
| **Gender** | | | p<0,001* |
| Male | 63 (21,4) | 1.344 (42,9) | |
| Female | 231 (78,6) | 1.790 (57,1) | |
| **Race/ethnicity** | | | 0,242 |
| Black | 71 (24,4) | 649 (21,0) | |
| Brown | 84 (28,9) | 874 (28,2) | |
| White | 132 (45,4) | 1.482 (47,9) | |
| Asian/Indigenous | 4 (1,4) | 91 (2,9) | |
| **Marital status** | | | 0,011 |
| Married/Common-law marriage | 175 (59,5) | 2.135 (68,1) | |
| Separated/Divorced/Widowed | 82 (27,9) | 684 (21,8) | |
| Single | 37 (12,6) | 315 (10,1) | |
| **Education** | | | p<0,001* |
| Primary education | 16 (5,4) | 522 (16,7) | |
| Secondary education | 124 (42,2) | 1.219 (38,9) | |
| Higher education | 154 (52,4) | 1.393 (44,4) | |
| **Occupational category** | | | 0,007 |
| Higher | 95 (32,8) | 980 (31,8) | |
| Middle | 156 (53,8) | 1.452 (47,2) | |
| Manual | 39 (13,4) | 646 (21,0) | |
| **Monthly per capita Income (BR reais)** | 1.680,9±1.425,6 | 1.563,9±1.330,7 | 0,116 |
| **Smoking habit** | | | 0,001 |
| Never smoked | 190 (64,6) | 1.691 (54,0) | |
| Former smoker | 73 (24,8) | 1.102 (35,2) | |
| Current smoker | 31 (10,5) | 341 (10,9) | |
| **Alcohol use** | | | 0,690 |
| Non-user | 105 (35,7) | 1.082 (34,6) | |
| User | 189 (64,3) | 2.049 (65,4) | |
| **Leisure-time PA** | | | 0,347 |
| Weak | 235 (81,6) | 2.576 (83,1) | |
| Moderate | 37 (12,8) | 405 (13,1) | |
| Strong | 16 (5,6) | 118 (3,8) | |
| **Bariatric surgery** | | | p<0,001* |
| No | 273 (93,8) | 3.068 (98,8) | |
| Yes | 18 (6,2) | 38 (1,2) | |
| **BMI (kg/m²)** | 32,9±2,9 | 33,8±3,6 | p<0,001* |
| **WC (cm)** | 100,5±8,6 | 106,9±10,1 | p<0,001* |

*(Continued)*

**Table 2.** (Continued)

| Variables | Baseline | | |
|---|---|---|---|
| | **MHO** | **MUO** | **p-Value** |
| | **294 (8,6)** | **3.134 (91,4)** | |
| % of grams of MPF W1 and W3 | 76,2±9,5 | 76,9±10,5 | 0,157 |
| % of grams of PF W1 and W3 | 9,1±6,2 | 9,9±7,6 | 0,710 |
| % of grams of UPF W1 and W3 | 14,6±7,8 | 13,2±8,0 | **0,002** |

**Note:** PA: physical activity; BMI: Body Mass Index; WC: waist circumference; MPF: minimally processed food; PF: processed food; UPF: ultra-processed food. W1: Wave 1or baseline; W3: Wave 3.Variables were expressed as mean±standard deviation or n (%). Student's t-test and Mann-Whitney test were used.

The variables from Table 3 with $p < 0.20$ were included as potential predictors of metabolic status transition risk. Thus, the model included sex ($p = 0.167$), alcohol use ($p = 0.007$), waist circumference ($p = 0.002$), % of AMP ($p = 0.123$), and % of AUP ($p = 0.108$). The crude model refers to the risk analysis between each described variable and the metabolic status transition variable. In the adjusted model, all variables from the table were included in the regression and were related to the metabolic status transition variable.

The relative risk analysis among the variables is described in Table 4. After adjustments, individuals who consume alcohol showed a greater risk for metabolic status transition [RR: 1.238 (95% CI: 1.004–1.499)]. Additionally, a 1 cm increase in WC contributes to a 1% increase in the risk of changing from healthy to unhealthy metabolic status [RR: 1.011 (95% CI: 1.004–1.018)].

Furthermore, sensitivity analyses were conducted regarding a few variables used. Regarding physical activity (PA), in addition to the categorical variable used in the study, we also tested PA as a continuous variable considering the duration of strong, weak, and moderate PA per week, but none showed statistically significant results. Also, as a measure of quality control, sensitivity analysis was performed by three independent researchers (FDM; CPF; HCS) to clarify doubts regarding the classification of certain foods as processed or ultra-processed: feijoada, popcorn, stroganoff, yakisoba, coffee with sweetener, natural juice with sweetener, and tea/mate with sweetener. There was no statistical difference regarding the two ways of classifying the foods; therefore, it was decided to display the results according to the classification previously described in the methodology and used in other ELSA-Brasil's publications.

Regarding the sensitivity analysis, the variables that showed significant differences in the methods of Wildman et al. and Quintino were similar, so we opted to use the stricter criteria proposed by the later in this study [9,16]. Similarly, the data analysis with and without the individuals who underwent bariatric surgery was comparable, so we chose to include them in the study. According to the dietary assessment, there was no difference presented between the approach using grams or calories, so we used grams in this study in order to keep diet/light beverages in the analysis. Additionally, there was no statistical difference regarding the reclassification of doubtful foods in the NOVA system. The continuous physical activity variables also did not show statistical differences between the groups, so we used the categorical variable in this work. The diet change variable had a p-value of 0.339, indicating that this is not a factor influencing the findings of the study.

Also, the final regression was tested using the follow up duration as a potential confounder and it yielded extremely similar results, demonstration no significant effect on the risk factors identified or their measures of association. Finally, the analysis of losses was conducted, and there was no statistical difference in any of the variables, thus concluding that the losses were non-differential.

## Discussion

Our work indicates that most MHO individuals evolve into MUO after an average 7.7 years, reinforcing that MHO is a transitional state. Furthermore, alcohol appears to stimulate the process of metabolic status transition. Moreover, an

**Table 3. Characterization of metabolic status transition according to sociodemographic, lifestyle, anthropometric, and dietary consumption variables after an average 7.7 years of follow-up in the ELSA-Brasil study.**

| | Metabolic status transition between baseline and follow-up | | |
|---|---|---|---|
| | No | Yes | p-Value |
| | **46 (24,2)** | **144 (75,8)** | |
| **Age** | | | 0,928 |
| Quartile 1(≤ 53.0 years) | 14 (30,4) | 39 (27,1) | |
| Quartile 2(54.0–58.0 years) | 10 (21,7) | 35 (24,3) | |
| Quartile 3(59.0–65.0 years) | 11 (23,9) | 39 (27,1) | |
| Quartile 4(≥ 66.0 years) | 11 (23,9) | 31 (21,5) | |
| **Gender** | | | 0,167 |
| Male | 7 (15,2) | 36 (25,0) | |
| Female | 39 (84,8) | 108 (75,0) | |
| **Race/ethnicity** | | | 0,558 |
| Black | 10 (21,7) | 32 (22,5) | |
| Brown | 18 (39,1) | 42 (29,6) | |
| White | 18 (39,1) | 66 (46,5) | |
| Asian/Indigenous | 0 (0,0) | 2 (1,4) | |
| **Marital status** | | | 0,304 |
| Married/Common-law marriage | 23 (50,0) | 90 (62,5) | |
| Separated/Divorced/Widowed | 14 (30,4) | 35 (24,3) | |
| Single | 9 (19,6) | 19 (13,2) | |
| **Education** | | | 0,643 |
| Primary education | 1 (2,2) | 8 (5,6) | |
| Secondary education | 17 (37,0) | 51 (35,4) | |
| Higher education | 28 (60,9) | 85 (59,0) | |
| **Occupational category** | | | 0,530 |
| Higher | 11 (25,0) | 48 (33,8) | |
| Middle | 27 (61,4) | 75 (52,8) | |
| Manual | 6 (13,6) | 19 (13,4) | |
| **Monthly per capita Income (BR reais)** | 3.941,2±3.342,6 | 3.449,4±2.707,9 | 0,427 |
| **Smoking habit** | | | 0,689 |
| Never smoked | 30 (65,2) | 86 (60,1) | |
| Former smoker | 14 (30,4) | 46 (32,2) | |
| Current smoker | 2 (4,33,4) | 11 (7,7) | |
| **Alcohol use** | | | **0,007** |
| Non-user | 27 (58,7) | 52 (36,1) | |
| User | 19 (41,3) | 92 (63,9) | |
| **Leisure-time PA** | | | 0,481 |
| Weak | 34 (73,9) | 111 (77,1) | |
| Moderate | 10 (21,7) | 22 (15,3) | |
| Strong | 2 (4,3) | 11 (7,6) | |
| **Bariatric surgery** | | | 0,231 |
| No | 42 (91,3) | 138 (95,8) | |
| Yes | 4 (8,7) | 6 (4,2) | |
| **BMI (kg/m²)** | 34,3±2,5 | 35,0±3,5 | 0,332 |
| **WC (cm)** | 103,9±9,0 | 109,7±10,6 | **0,002** |
| **Dietary change** | | | 0,339 |

*(Continued)*

**Table 3.** (Continued)

| | Metabolic status transition between baseline and follow-up | | |
| --- | --- | --- | --- |
| | No | Yes | p-Value |
| | **46 (24,2)** | **144 (75,8)** | |
| No | 23 (50,0) | 83 (58,0) | |
| Yes | 23 (50,0) | 60 (42,0) | |
| % of grams of MPF W1 and W3 | 78,1±7,5 | 75,5±10,0 | 0,123 |
| % of grams of PF W1 and W3 | 9,0±5,7 | 9,4±6,1 | 0,785 |
| % of grams of UPF W1 and W3 | 12,9±6,1 | 15,1±8,2 | 0,108 |

**Note:** PA: physical activity; BMI: Body Mass Index; WC: waist circumference; MPF: minimally processed food; PF: processed food; UPF: ultra-processed food.W1: Wave 1or baseline; W3: Wave 3. Variables were expressed as mean±standard deviation or n (%). Student's t-test and Mann-Whitney test were used.

increase in waist circumference also is a significant predictor for the risk of transitioning from a healthy metabolic status to an unhealthy one.

The data shows that 75.8% of the MHO individuals at baseline experienced a transition in metabolic status, moving from healthy to unhealthy after follow-up. Similar findings were reported by researchers who used data from the Framingham Heart Study, where 71.3% of the 230 MHO individuals transitioned to MUO after nearly 13 years of monitoring [1]. Furthermore, a study conducted at a private hospital in São Paulo demonstrated that out of the 812 participants with MHO at the beginning of the study, 35.3% underwent a transition in status after 3.5 years of follow-up, becoming metabolically unhealthy individuals with obesity (2). The work of Palatini and colleagues, conducted with a population from Italy (n = 1,210), indicated that the MHO state is an unstable condition, as after 7.5 years of follow-up, 59.3% of individuals with MHO transitioned to an unhealthy status [5].

Data from a literature review and a study conducted in China with 458,246 adults showed that potential mediators of the transition in metabolic status include body fat distribution, with the accumulation of visceral fat and impaired adipose tissue function, resulting in hypertrophy and impaired lipid metabolism, as well as inflammation and adipogenesis [7,27]. These changes contribute to a higher burden of reactive oxygen species, which leads to increased lipid peroxidation, decreased adiponectin, damage to adipocyte DNA, and lipogenesis [27,28]. Although the MHO condition is considered relatively healthy, some authors argue that it cannot be regarded as a state of total health, as reactive oxygen species produced in obesity cause chronic damage to lipoproteins [28]. Thus, when a cardiometabolic alteration described in the MHO diagnostic criteria occurs, the transition of metabolic status happens [4].

It is believed that metabolic health plays a fundamental role in predicting adverse health risks over the years [4]. Thus, those who undergo a transition in metabolic status may develop changes in glycemic and lipid profiles, as well as blood pressure, which can result in the incidence of type II diabetes, hypertension, dyslipidemias, and culminate in other associated diseases such as cardiovascular diseases, chronic kidney disease, and cancer [11,13]. Concurrently, after transitioning to an unhealthy status, the individual should exhibit a higher risk of all-cause mortality [29].

It was noted, then, that individuals who are alcohol consumers are at greater risk for the transition in metabolic status. Corroborating this finding, a study conducted on 3,669 Taiwanese military personnel showed that among the MHO individuals, 55.8% (n = 181) did not consume alcohol, compared to 53.2% (n = 647) of the MUO individuals [10]. Furthermore, a study conducted amid 728 people with alcohol use disorders in Spain showed that the prevalence of metabolic syndrome was 13.9% (n = 101). Among these patients, 86.9% were individuals with obesity, 27.7% determined as having diabetes, and 78.2% had elevated fasting glucose levels. Additionally, 70.3% had high blood pressure, 15.3% had a history of cardiovascular disease, 80.2% had elevated triglyceride levels, and 67.3% had low HDL cholesterol levels [30].

It is known that excessive alcohol consumption is toxic to all tissues and bodily systems. A study conducted in Russia with 2,381 adults showed that alcohol consumption influences energy intake by increasing appetite and affecting satiety, thus promoting weight gain [31]. Furthermore, through its impact on the sympathetic nervous system and the release of nitric oxide, excessive alcohol consumption impairs endothelial vasodilatation, leading to endothelial dysfunction and cell apoptosis, contributing to the development of hypertension [32]. Moreover, it is suggested that the effects of alcohol on cardiometabolic alterations are influenced by its impact on WC [31,32]. Contributing to this statement, a study conducted with 2,629 adults from Russia classified individuals based on alcohol consumption into four groups: non-drinkers, non-problematic drinkers, hazardous drinkers, and harmful drinkers, with the first group having zero alcohol consumption and the last group exhibiting the highest consumption. The study showed that hazardous-drinking men had a larger WC of 6.11 cm than non-problematic drinkers, whereas non-problematic drinking women had a larger WC of 2.92 cm when compared to non-drinkers [31].

WC by itself was shown to be associated with a change in metabolic status, as our study confirmed that a 1 cm increase in WC increases the risk of transitioning from MHO to MUO by 1%. Reinforcing this statement, the work of Hamer and colleagues does not attribute the transition of status to lifestyle habits such as physical activity and diet, but rather to increases in waist circumference, which possibly reflects changes in visceral adiposity [8]. Furthermore, a study conducted with 13,525 Chinese adults showed that individuals classified as metabolically healthy had a smaller WC compared to those classified as unhealthy [33].

It is believed that the relationship between WC and metabolic status is justified by its impact on the components of metabolic syndrome. Excess abdominal adiposity leads to overstimulation of leptin secretion and increases sodium absorption after activation of the renin-angiotensin-aldosterone system, resulting in increased blood pressure [34]. Furthermore, the chronic state of inflammation, characterized by increased levels of C-reactive protein, inflammatory cytokines, resistin, leptin, and adiponectin, affects insulin resistance, leading to the exhaustion of pancreatic beta cells and impairing the maintenance of normoglycemia [10,34].

Sex did not demonstrate being associated with the risk of metabolic status transition. A similar result was found by Palatini and colleagues, indicating that sex was not a significant predictor for metabolic status transition [5]. Another study conducted with participants from the United States carried out a 30-year follow-up to observe the transition of metabolic

**Table 4. Relative Risk and Confidence Interval for the Transition of Metabolic Status in Follow-up and Sociodemographic, Lifestyle, Anthropometric, and Dietary Consumption Variables from ELSA-Brasil.**

| Variables | Raw model | Adjusted Model |
|---|---|---|
| | RR (95% CI) | RR (95% CI) |
| **Gender** | | |
| Female | Ref | Ref |
| Male | 1,140 (0,967–1,342) | 1,010 (0,854 -1,194) |
| **Alcohol use** | | |
| Non-user | Ref | Ref |
| User | **1,259 (1,052–1,507)** | **1,238 (1,023–1,499)** |
| **WC (cm)** | **1,012 (1,005—1,019)** | **1,011 (1,004–1,018)** |
| **% of grams of MPF W1 and W3** | 0,993 (0,986–1,000) | 1,006 (0,994–1,018) |
| **% of grams of UPF W1 and W3** | 1,008 (1,000–1,017) | 1,012 (0,998–1,026) |

**Note**: RR: Relative Risk; CI: Confidence Interval; BMI: Body Mass Index; WC: Waist Circumference; MPF W1 and W3: average percentage of minimally processed foods at wave 1 and 3.UPF W1 and W3: average percentage of ultra-processed foods at wave 1 and 3. Poisson regression–robust estimator. Model I: Adjusted for all included variables.

status and stated that women move in and out of the MHO classification over the years, while men do not; after transitioning from MHO to MUO status, they generally remain in the unhealthy state [6].

Regarding eating habits, two longitudinal studies—one conducted in Spain with 5,373 older adults and another in Brazil with 8,065 adults and older adults—showed an association between diet, particularly its (higher) content of ultraprocessed foods, and components of metabolic syndrome [14,35]. However, in our study the consumption of MP and UP did not appear to be associated with the risk of metabolic status transition. Corroborating this finding, a study conducted with 160 workers in Brazil showed that the NOVA classification is not associated with metabolic status [3]. The dietary intake of ELSA participants shows a higher percentage of MP consumption in both groups, MHO and MUO (78.1% and 75.5%, respectively), with no statistical difference between them. This value is higher than the average value for the Brazilian population, which consumes an average of 53.4% of natural and minimally processed foods [36]. It seems that the individuals analyzed in ELSA have better dietary habits than the whole of the Brazilian population, which may have made it difficult to find any association between diet and metabolic status.

It is worth noting that the study has some limitations. The lack of consensus on the criterion for evaluating MHO hinders comparison with other studies. Additionally, the ELSA-Brasil sample is not representative of the Brazilian population. Furthermore, participants' dietary patterns are similar, which might influence the obtained results; not to mention that the variable is subject to participant memory and response bias, which can underestimate or overestimate consumption. However, the study's strengths are highlighted: the longitudinal design allows establishing a cause-and-effect relationship between the analyzed variables. The criterion used to assess WHO does not consider any cardiometabolic alterations, ensuring that individuals are indeed healthy at the time of diagnosis. Despite the self-reported consumption data, the questionnaires were validated for the Brazilian population, and the interviewers were previously trained, ensuring the internal validity of the data.

## Conclusion

The metabolic health of most individuals with obesity appears to be a dynamic state, as evidenced by the significant proportion of participants classified as metabolically healthy obese (MHO) who transitioned to the metabolically unhealthy obese (MUO) state during the follow-up period. Our findings align with previous research, emphasizing the vulnerability of the MHO phenotype to metabolic decline, as only a small minority were able to maintain metabolic health over the years.

The analysis reveals that visceral fat accumulation is a key factor driving the transition from MHO to MUO, with alcohol consumption also contributing to this metabolic decline. These insights highlight critical opportunities for interventions aimed at preventing or delaying this transition, such as promoting the reduction of visceral fat and encouraging healthier lifestyle choices.

In conclusion, while some individuals with obesity may appear metabolically healthy for a time, it is important to recognize that obesity remains a significant long-term health risk. All individuals with obesity, regardless of their immediate metabolic status, should receive appropriate and ongoing care that focuses on overall health and wellbeing rather than just weight loss. Even those who are temporarily considered healthy still face a significant risk of metabolic decline, emphasizing the need for ongoing and effective care. This requires a holistic approach that goes beyond just eating habits and physical activity to also include alcohol consumption and overall health management. This approach is essential not only to reduce the prevalence of obesity-related complications such as diabetes and hypertension but also to support the long-term well-being and quality of life for individuals living with obesity.

## Author contributions

**Conceptualization:** Fernanda Duarte Mendes, Carolina Perim de Faria.

**Formal analysis:** Fernanda Duarte Mendes, Carolina Perim de Faria.

**Funding acquisition:** José Geraldo Mill, Maria Del Carmen Bisi Molina.

**Investigation:** Fernanda Duarte Mendes, José Geraldo Mill, Maria Del Carmen Bisi Molina.

**Project administration:** José Geraldo Mill, Maria Del Carmen Bisi Molina.

**Resources:** José Geraldo Mill.

**Supervision:** Carolina Perim de Faria.

**Visualization:** Hully Cantão dos Santos, Maria de Fátima H Sander Diniz, Carla Romagnolli Quintino, Márcio Sommer Bittencourt, Carolina Perim de Faria.

**Writing – original draft:** Fernanda Duarte Mendes, Carolina Perim de Faria.

**Writing – review & editing:** Hully Cantão dos Santos, José Geraldo Mill, Maria Del Carmen Bisi Molina, Maria de Fátima H Sander Diniz, Carla Romagnolli Quintino, Márcio Sommer Bittencourt.

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
