## [Decision Letter · Decision Letter 0]

14 Jan 2025

PGPH-D-24-02621

From Healthy to Unhealthy Obesity: A Longitudinal Study of Adults in ELSA-Brasil

Dear Dr. Faria,

Thank you for submitting your manuscript to PLOS Global Public Health. After careful consideration, we feel that it has merit but does not fully meet PLOS Global Public Health’s publication criteria as it currently stands. Therefore, we invite you to submit a revised version of the manuscript that addresses the points raised during the review process.

We look forward to receiving your revised manuscript.

Kind regards,

Vanessa Moraes Bezerra

Academic Editor

Journal Requirements:

2. In the online submission form, you indicated that "Data described in the manuscript, code book, and analytic code will be made available upon reasonable request.". 

a. In a public repository, 

b. Within the manuscript itself, or 

c. Uploaded as supplementary information.

Additional Editor Comments (if provided):

Reviewers' comments:

Reviewer's Responses to Questions

**Comments to the Author**

1. Does this manuscript meet PLOS Global Public Health’s publication criteria ? Is the manuscript technically sound, and do the data support the conclusions? The manuscript must describe methodologically and ethically rigorous research with conclusions that are appropriately drawn based on the data presented.

Reviewer #1: Yes

Reviewer #2: Yes

2. Has the statistical analysis been performed appropriately and rigorously?

Reviewer #1: Yes

Reviewer #2: Yes

3. Have the authors made all data underlying the findings in their manuscript fully available (please refer to the Data Availability Statement at the start of the manuscript PDF file)?

Reviewer #1: Yes

Reviewer #2: Yes

4. Is the manuscript presented in an intelligible fashion and written in standard English?

Reviewer #1: Yes

Reviewer #2: Yes

5. Review Comments to the Author

Reviewer #1: e MHO as a transitory state, which could be revised to reflect the limitations of current knowledge on the subject.

Regarding the discussion, I recommend including reflections on the challenges of caring for individuals with obesity, highlighting the importance of overcoming practices focused exclusively on weight loss and the pathologization of larger bodies. Considering obesity as a chronic and recurrent condition, it is crucial to emphasize empathetic and comprehensive care strategies that promote healthy lifestyles and embrace individual experiences. These adjustments could significantly contribute to enhancing the relevance and applicability of the results, both in the clinical context and in the development of more inclusive health policies.

Introduction

In the second paragraph of the introduction, the current wording suggests that the authors already assert the nonexistence of metabolically healthy obesity (MHO). I recommend revising this section to provide a balanced discussion of the evidence both supporting and opposing this condition (lines 48-62, page 7).

Additionally, the phrasing of the hypothesis implies that the group assumes MHO to be a transitional state. Given the current limitations in knowledge on this topic, I suggest rethinking how this hypothesis is presented to better reflect the uncertainties and nuances still under investigation.

Methods

What is the reason for not using data from the second wave to track weight variation over the years?

Overall, the methods are well-described and meet the requirements of the proposed analyses, but including this information could enhance the understanding of the approach adopted.

Results

I would like to understand why the variables associated with changes in body weight over the years were not presented. Were these data analyzed? If so, did they have a relationship with MHO?

I believe this analysis would be crucial to better understand the relationship between obesity diagnosed by BMI and the observed metabolic changes.

Discussion

In line 325, I suggest paying attention to the use of acronyms and recommend replacing them with full terms for better clarity.

Overall, the discussion is well-grounded and incorporates important references on the topic. However, it would be valuable to include reflections on the challenges of caring for people with obesity and the experience of living with a chronic and recurrent condition. It is urgent to construct new narratives that help move beyond practices focused exclusively on weight loss and the pathologization of larger bodies.

I invite the authors to reflect on the importance of care strategies that do not reinforce blame and, instead, promote the development of more empathetic and inclusive practices that support the maintenance of a healthy life regardless of weight loss. Metabolically healthy obesity is a complex issue that can contribute to the development of broader and health-centered indicators rather than relying solely on weight or the exclusive use of BMI as a monitoring parameter.

Conclusion Revision

The evidence found highlights alcohol consumption and increased waist circumference as risk factors for metabolic transition. This underscores the need to promote healthy habits as central components of obesity care.

Considering obesity as a chronic and recurrent condition, we emphasize that care must go beyond an exclusive focus on weight loss. It is essential to incorporate balanced eating, regular physical activity, moderation in alcohol consumption, stress management, and other aspects. These measures can contribute to preventing metabolic changes and promoting overall well-being.

Reviewer #2: Thank you for the opportunity to review this paper. This article is relevant to health promotion and prevention of obesity-related complications.

The methodology used in this study is suitable for answering the research objectives. There are issues that could be best described:

Methods

Line 112: "Food intake was analyzed longitudinally (dynamic model),...".

The authors need to better explain or reference the model used to estimate food intake

Line 116 - 119: "The International Physical Activity Questionnaire (IPAQ) long version, validated for Brazil by Matsudo et al. (23), was applied. Physical activity was recorded in minutes per week and subdivided into low, moderate, and high categories, according to the World Health Organization (24)."

It is not clear how the physical activity variable was calculated and categorized. What are the cutoff points? It is necessary to justify the choice of the term low, moderate, and high.

The studies, for the most part, classify whether the individual is inactive, insufficiently active or active

Results

Age was divided into quartiles. Report the cutoff points for each category

In the tables, inform in the legend what O1 and O3 of the variables mean: % of grams of MPF O1 and O3, % of grams of PF O1 and O3 and % of grams of UPF O1 and O3

Line 244-246: "Also, the final regression was tested using the follow up duration as a potential confounder and it yielded extremely similar results, demonstration no significant effect on the risk factors identified or their measures of association."

The authors need to explain why they considered follow-up time as a confounding factor. By considering follow-up time in the analysis, the measure of occurrence and association would change.

Discussion

It would be easier for the reader if the authors, when comparing with the literature, informed the study population and location of the cited reference. For example:

Line 261-262: "The work of Palatini and colleagues indicated that the MHO state is an unstable condition, as after 7.5 years of follow up, 59.3% of MHO individuals transitioned to an unhealthy status (5)."

Line 286-287: "Among these patients, 86.9% were individuals with 287 obesity, 27.7% determined as having diabetes, and 78.2% had elevated fasting glucose levels."

Line 319-322: "Another study conducted a 30-year follow-up to observe the 320 transition of metabolic status and stated that women move in and out of the MHO classification over the years, while men do not; after transitioning from MHO to MUO status, they generally remain in the unhealthy state(6)."

6. PLOS authors have the option to publish the peer review history of their article (what does this mean? ). If published, this will include your full peer review and any attached files.

**Do you want your identity to be public for this peer review?** For information about this choice, including consent withdrawal, please see our Privacy Policy .

Reviewer #1: **Yes: ** Poliana Cardoso Martins

Reviewer #2: No

---

## [Decision Letter · Decision Letter 1]

4 Apr 2025

From Healthy to Unhealthy Obesity: A Longitudinal Study of Adults in ELSA-Brasil

PGPH-D-24-02621R1

Dear Prof. Faria,

We are pleased to inform you that your manuscript 'From Healthy to Unhealthy Obesity: A Longitudinal Study of Adults in ELSA-Brasil' has been provisionally accepted for publication in PLOS Global Public Health.

Best regards,

Julia Robinson

Executive Editor

Reviewer Comments (if any, and for reference):

Reviewer's Responses to Questions

**Comments to the Author**

1. If the authors have adequately addressed your comments raised in a previous round of review and you feel that this manuscript is now acceptable for publication, you may indicate that here to bypass the “Comments to the Author” section, enter your conflict of interest statement in the “Confidential to Editor” section, and submit your "Accept" recommendation.

Reviewer #2: All comments have been addressed

2. Does this manuscript meet PLOS Global Public Health’s publication criteria ? Is the manuscript technically sound, and do the data support the conclusions? The manuscript must describe methodologically and ethically rigorous research with conclusions that are appropriately drawn based on the data presented.

Reviewer #2: Yes

3. Has the statistical analysis been performed appropriately and rigorously?

Reviewer #2: Yes

4. Have the authors made all data underlying the findings in their manuscript fully available (please refer to the Data Availability Statement at the start of the manuscript PDF file)?

Reviewer #2: Yes

5. Is the manuscript presented in an intelligible fashion and written in standard English?

Reviewer #2: Yes

6. Review Comments to the Author

Reviewer #2: (No Response)

7. PLOS authors have the option to publish the peer review history of their article (what does this mean? ). If published, this will include your full peer review and any attached files.

**Do you want your identity to be public for this peer review?** For information about this choice, including consent withdrawal, please see our Privacy Policy .

Reviewer #2: No
